# Attribution Preservation in Network Compression for Reliable Network Interpretation

**Geondo Park**[1,*] **June Yong Yang**[1,*] **Sung Ju Hwang**[1,2], **Eunho Yang**[1,2]
KAIST[1], AITRICS[2], South Korea
{geondopark, laoconeth, sjhwang82, eunhoy}@kaist.ac.kr

## Abstract

Neural networks embedded in safety-sensitive applications such as self-driving cars and wearable health monitors rely on two important techniques: input attribution for hindsight analysis and network compression to reduce its size for edge-computing. In this paper, we show that these seemingly unrelated techniques conflict with each other as network compression deforms the produced attributions, which could lead to dire consequences for mission-critical applications. This phenomenon arises due to the fact that conventional network compression methods only preserve the predictions of the network while ignoring the quality of the attributions. To combat the attribution inconsistency problem, we present a framework that can preserve the attributions while compressing a network. By employing the Weighted Collapsed Attribution Matching regularizer, we match the attribution maps of the network being compressed to its pre-compression former self. We demonstrate the effectiveness of our algorithm both quantitatively and qualitatively on diverse compression methods.

## 1 Introduction

Riding on the recent success of deep learning in numerous fields, there is an emergent trend to utilize deep neural networks (DNNs) even for safety-critical applications such as self-driving cars and wearable health monitors. Due to the inherent nature of such devices, it is of paramount importance that the utilized DNNs be *reliable* and *trustworthy* to human users.

For a system to be reliable, perpetual service must be rendered and the integrity of the system must hold even under unexpected circumstances. For most commercially deployed DNNs, this condition is hardly met as they are often operated in the cloud due to their heavy computational requirements. However, this dependence on clouds acts as a critical weakness in safety-sensitive settings as intermittent communication failures to the cloud may cause difficulties in reacting to situations immediately, or even worse, the device's connection to the cloud may be severed indefinitely. Thus, to guarantee reliable service, the DNNs must be embedded on the edge device. To this end, network compression techniques such as pruning [1, 2] and distillation [3, 4] are commonly employed - as a compressed network would require less computational time and memory but maintain its prediction performance to a certain acceptable margin, effectively substituting the original network for edge computation.

At the same time, for a system to be trustworthy, the system must be transparent enough for humans to understand its workings and the reasons for its outputs. An example would be when a health monitor predicts an onset of a disease [5] - then the clinician would require an acceptable explanation to the device output. However, the black-box nature of deep neural networks complicates this goal - impeding its advance in safety-critical areas. For DNNs to gain trustworthiness, the ability to explain

---

[*]Equal contribution. Listing order is alphabetical.

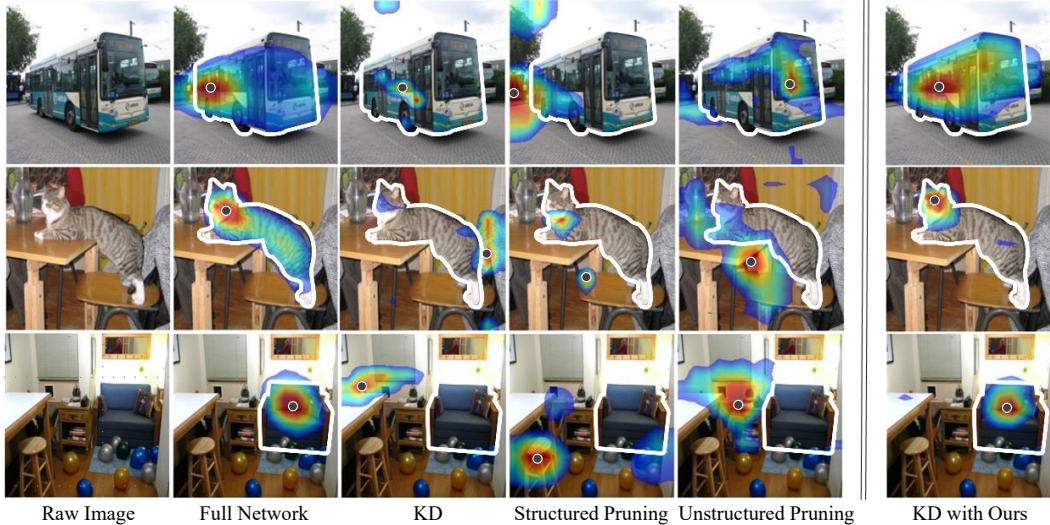

| Raw Image | Full Network | KD | Structured Pruning | Unstructured Pruning | KD with Ours |

**Figure 1:** Attribution maps of a network before and after network compression. These figures are examples that the networks before and after compression predicted the same correct labels (bus, cat, sofa), but exhibit different attribution maps. Observe that for compressed networks, the max value of the heatmaps (blue circle) is evicted outside the segmentation boundaries (white line) while our method maintains the dot.

why the network makes such decisions is essential. Such field of interest - eXplainable AI (XAI) - has emerged as one of the important frontiers in the field of deep learning. Among numerous XAI methods, the most commonly used methods are attribution methods [6], which weigh the parts of the input data according to how much they 'contributed' to produce the output prediction. Such attribution methods are beginning to be applied in safety-critical fields [7].

To ensure the safety of the system, the two aforementioned conditions should be simultaneously satisfied - the embedded DNNs must be equipped with both compression and attribution. However, we show for the first time that these seemingly unrelated techniques conflict with each other: compressing a network causes deformations in the produced attributions, even if the predictions of the network stays the same before and after compression (See Figure 1). This is a potentially severe crack in the integrity of the compressed network, as the premise in which a compressed network is acceptable in safety-critical fields is that the compressed network is as *reliable* as its former self. This implies that the compressed network must behave almost identically to the pre-compression network while being smaller in size. Moreover, the attributions of the compressed network are not only different from their past counterparts but also broken down compared to their respective segmentation ground truths, as shown in Figure 1 and Table 1. These attribution distortions directly cause incorrect interpretations, which could lead to dire consequences for safety-critical systems. Such a problem arises from the pitfall of existing network compression approaches: they only aim to maintain the prediction quality of the network while reducing the size of the network.

Compressing a network forces the network to cram its necessary decision procedures and information inside a smaller space. This space restriction forces the network to abandon its standard decision procedures and resort to using shortcuts and hints that are seemingly indecipherable to humans. Thus, its decision procedures would become harder to interpret, which is reflected in its production of deformed attribution maps.

To resolve this newfound unintended issue, we propose a novel attribution-aware compression framework to ensure

**Table 1:** Evaluation of how many samples were broken compared to the ground truth (segmentation labels) by various compression methods. Here, AUC denotes the degree of overlap between the segmentation and attribution map (see Section 4). Point accuracy [8] is a measure of whether the max value of the heatmap is inside the segmentation map. Only the samples that the predictions of the network were **correct** are counted.

|  | For samples with correct pred. | | |
| Method | #Param | AUC | Point Acc |
| --- | --- | --- | --- |
| Full (Teacher) | 15.22M | 88.79 | 80.21 |
| Knowledge Distillation | 0.29M | 78.74 | 67.26 |
| Structured Pruning | 3.27M | 79.98 | 75.29 |
| Unstructured Pruning | 0.53M | 84.13 | 75.43 |
| KD (w/ Ours) | 0.29M | 88.06 | 79.12 |

both the reliability and trustworthiness of the compressed model. One way to tackle this problem is to inject the attribution information to the now-compressing network by employing a matching regularizer to match the attributions to a ground truth signal (e.g. ground truth segmentation data). However, these kinds of signals are very rare as they require extensive human labor. To bypass this problem, we concentrate on the observation that the attributions of the pre-network (teacher) are closer to the ground truth signal compared to the post-network (student), as shown in Table 1. Thus, in the absence of ground truth signals, the attributions of the teacher can serve as a proxy. In this sense, we propose a regularizer that matches the attribution maps of the now-compressing network to its attribution maps before compression, transferring the attributional power of the pre-network to the post-network. Our work sheds new light on transfer learning techniques from the perspective of XAI, as they can be re-interpreted and subsumed under our framework.

Our contributions are as follows:

- We show for the first time that compressing networks via pruning or distillation distorts the attributions of the network (i.e. compressed networks classify correctly but pay attention to *wrong* places), hence the well-calibrated explainability of the original model can be completely destroyed even with matched performance.

- We propose a matching technique to efficiently preserve diverse levels of attribution maps while compressing the networks, by regularizing the differences between the sampled attribution maps of the teacher and the student.

- Through extensive experiments, we validate the effectiveness of our framework and show that our attribution matching not only maintains the interpretation of the model but also yields significant performance gains.

## 2   Related Work

**Attribution Methods**   Recent advances in producing human-understandable explanations for predictions of DNNs have gained much attention throughout the machine learning community. Among a variety of approaches towards this goal, one widely adopted method of interpretation is input attribution. Attribution approaches try to explain deep neural networks by producing *visual explanations* about the decisions of the network. By examining how the network's output reacts to change in the input, the contributions of each input variable are calculated. In computer vision, these contributions are displayed in a 2-D manner, forming an attribution map. Attribution maps identify the spatial locations of the parts of the image the network deems significant in producing such a decision. Early works toward this direction use the gradient of the network output with respect to the input pixels to represent the sensitivity and significance of specific input pixels [9, 10, 11]. More recent studies such as Guided Backprop [12], Grad-Cam [6] or integrated gradients [13] proposed to process and combine these gradient signals in more careful ways. Another line of works proposed to propagate relevance values in a way that their total amount is preserved for a single layer. These relevance scores are backpropagated through the network from the output layer to the input layer. Several studies such as EBP [14], LRP [15] proposed to define novel relevance scores differing from vanilla gradients and backpropagate these values according to a set of novel backpropagation rules.

**Network compression**   Commonly used deep neural networks are heavy in computation and memory by design. Their resource requirement is the main impediment in operating these networks on resource-constrained platforms. To alleviate this constraint, many branches of works have been proposed to reduce the size of an existing neural network. The most commonly employed approach is to reduce the number of weights, neurons, or layers in a network while maintaining approximately the same performance. This approach on deep neural networks was first explored in early works such as [16] and [17]. Recent studies conducted by [18, 1] has brought popularity to this line of work with a simple unstructured pruning method that reduces the size of the network by pruning unimportant connections within the network. However, unstructured pruning has an inherent weakness as it produces large sparse weight matrices that are computationally inefficient unless equipped with a specifically designed hardware. To resolve this issue, structured pruning methods were proposed [2, 19, 20] where entire channels are pruned simultaneously to ensure the denseness of the weights.

Network distillation, another branch of network compression initially proposed by [3], attempts to reduce the size of the network by transferring the knowledge of the full network to a student network

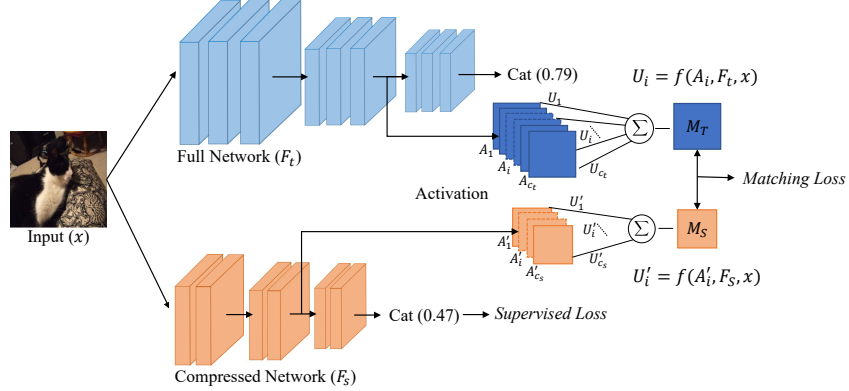

**Figure 2:** Overview of our framework

of smaller size. By employing a loss function that teaches the student network to mimic the outputs of the teacher network, a smaller network with similar performance can be obtained. Advanced methods of distillation have succeeded in achieving much more effective transfer by not only transferring the output logits but the information of the intermediate activations as in [4, 21, 22, 23].

# 3 Attribution-Preserving Compression

## 3.1 Background

**Network compression** Throughout the paper, we use the term network compression to refer to any activity that reduces the size of the network while maintaining the predictive performance of the network within a certain acceptable margin(pruning, distillation, quantization, and more). The general compression framework is composed of the following stages: Network pre-training, reduction, and fine-tuning. First, a full-size network $F_t$ (or teacher network) is trained. Next, the network is reduced in size. In the reduction phase, parts of the network (be it weights, channels, or information) are discarded, producing a network $F_s$ (or student network) that is smaller in size. For example, in the case of network pruning, the connections of the full network are severed with a pruning criterion and the according weights are discarded, shrinking the number of parameters in the network. Finally, the network $F_s$ is fine-tuned on the same dataset to sufficiently recover from the performance degradation caused by the reduction phase, producing the network $F_{s'}$. For certain kinds of algorithms such as network sparsification [20], steps two and three can be executed simultaneously.

**Attribution maps** For a neural network $F$, an attribution $M$ for an input data point $x$ at a certain layer is a multidimensional tensor containing the importance values of each input or neuron at that layer which the network considers important in making its according prediction. These attribution values are calculated based on the magnitude of the point and its sensitivity to change of value. Most attribution algorithms leverage the activation value (for magnitude) and gradient (for sensitivity) to determine the importance.

This definition can be readily applied to Convolutional Neural Networks (CNN). Consider a convolutional layer with kernel $K = \mathbb{R}^{k_h \times k_w \times c_{in} \times c_{out}}$ ($k_h$ and $k_w$ are the height and width of kernel respectively, and $c_{in}$ and $c_{out}$ are the number of input and output channels), and output activation $A = \mathbb{R}^{a_h \times a_w \times c_{out}}$. Then, the attribution of this layer is a 3-dimensional tensor $M \in \mathbb{R}^{a_h \times a_w \times c_{out}}$. However, due to the spatio-local nature of CNNs, the attributions are often summed and collapsed along their channel dimension to produce a spatial attribution map to enhance human-interpretability. Specifically, suppose we have an original 3-dimensional attribution $M'$ that is the concatenation of $C$ 2-dimensional (in $\mathbb{R}^{a_h \times a_w}$) attributions $\{M'_c\}_{c=1}^C$. Then, the collapsed version $M$ is computed as $\sum_{c=1}^C M'_c$.

## 3.2 Weighted Collapsed Attribution Matching Framework

We now present our framework, Weighted Collapsed Attribution Matching, which preserves the attributions in a compressed network by transferring the attributional power of its past self to the

current self. To this end, we employ a matching loss that matches the attribution map $M_t$ of $F_t$ to $M_s$ of $F_s$ in the fine-tuning stage of compression.

The key ingredient of our framework is the way of computing attribution maps. Beyond naively collapsing the 3-dimensional attribution to a 2-dimensional matrix, our framework allows to consider the importance of each channel when creating an attribution map. For the $l$-th layer of a CNN, the attribution map based on the importance-aware collapsing is produced in the following way:

$$M^{(l)} = V\bigg(\sum_{c=1}^{C} U_c^{(l)} \cdot T\big(A_c^{(l)}\big)\bigg) \tag{1}$$

where $A_c^{(l)}$ is output activation of channel $c$, $T(\cdot)$ is a function of choice, $U_c^{(l)} = f(A_c, F, x) \in \mathbb{R}$ is the importance of channel $c$ given by an importance calculation function $f$, and $V(\cdot)$ is an optional post-processing function. When it is clear from the context, we suppress the notation $(l)$ for clarity.

Given the weighted collapsed map in (1), we consider the following objective in the fine-tuning stage that tries to reduce the (normalized) $\ell_2$ difference between $M_t$ and $M_s$:

$$L_{total} = L(W_s, x) + \beta \sum_{j \in I} \left\| \frac{M_s^{(j)}}{\left\| M_s^{(j)} \right\|_2} - \frac{M_t^{(j)}}{\left\| M_t^{(j)} \right\|_2} \right\|_2 \tag{2}$$

where $L(W_s, x)$ is the supervised learning loss for $F_s$, $\beta$ is a tunable hyperparameter and $I$ is the set of layers to match. Note here that we use $\|\cdot\|_2$ to note element-wise $\ell_2$ norm (or Frobenius norm for matrix input). The overall schematic of our framework is depicted in Figure 2. For any kind of attribution algorithm that is end-to-end differentiable, we can directly apply and minimize our weighted collapsed attribution matching regularizer via stochastic gradient descent. This form of framework in (2) can be applied to any compression method that involves a fine-tuning phase - pruning, distillation, quantization, etc.

**Equally weighted collapsed activation map matching** A simple form of (1) is to naively assign equal importance weights to the channels and collapsing them along its channel dimension. Setting $U_c = 1$ for all $c$, $V(\cdot)$ and $T(\cdot)$ as element-wise identity and square function respectively, we have

$$M_{act} = \sum_{c=1}^{C} A_c^2. \tag{3}$$

where $A_c^2$ represents the Hadamard power (or element-wise power) of $A_c$. This regularizer was proposed in a prior work on transfer learning [4] to boost knowledge transfer from a teacher network to a student network, *just to improve performance*. This regularizer is viewed in the new light of XAI in our framework that it is matching label aggregated, channel-wise equally weighted attribution map. From our experiments below, we confirm that this regularizer is partially effective in preserving attribution maps in compression. However, this form of attribution map does not contain label-specific attribution information since all activation values are equally weighted and aggregated. In other words, this regularizer may teach the student how to look and distinguish objects, but does not pass on the information of 'what' and 'why' it should look at a certain region.

**Sensitivity-weighted activation map matching** As a practical showcase of our framework, we demonstrate a simple sensitivity-weighted matching regularizer. We elaborate on the flow of our framework using Grad-Cam, a simple yet effective and widely used attribution method. Grad-Cam produces an attribution map by aggregating the activation maps with a linear combination of activations, where each activation map is weighted by the sensitivity of the channel that is the label-specific pooled gradient of an activation map. Motivated by this, we define $U_c$ in (1) as

$$U_c = \sum_{a_h, a_w} \frac{\partial y^t}{\partial A_{(a_h, a_w, c)}} \tag{4}$$

where $y^t$ is the output logit generated by $F$ for some target class $t$. We set $V(\cdot)$ as ReLU to remove negative regions [6]. We also set $T(\cdot)$ as identity so that

$$M_{cg} = ReLU\bigg(\sum_{c=1}^{C} U_c \cdot A_c\bigg). \tag{5}$$

Unlike the collapsed activation map in (3), the activation maps are weighted by the pooled gradient values taken with respect to the output prediction. Since grad-cam is end-to-end differentiable, this form of regularization can be easily implemented within the conventional automatic differentiation framework. Since separate attribution maps can be created for each class label, we can match the attribution maps for all classes. However, to reduce the computational overhead, matching the attribution maps of high scoring classes is more plausible.

**Stochastic matching**    Another interesting family in framework (2) is the one leveraging stochasticity in computing importance weight $U_c$. Stochasticity injection in deep learning has been proven to exhibit generalization benefits [24, 25], and recent works are starting to utilize this concept to boost the performance of knowledge transfer between a teacher and a student [26, 27]. Inspired by these works, we formulate a stochastic matching regularizer to facilitate relevant information transfer and prevent overfitting in which the student network only learns to superficially imitate the attribution maps of the teacher. For this purpose, we impose a probability distribution in generating importance weights as $U_c \sim P_f(A_c, F, x)$ where $P_f$ is a probability distribution of the importance generating function $f$. In the fine-tuning phase, importance weights are sampled from the distribution and a perturbed attribution map is created.

A simple and applicable formulation is to impose a Bernoulli distribution on the importance weights. Specifically, similar to dropout, we draw i.i.d. samples from a Bernoulli distribution and mask the importance weights before summing the attributions. This is equivalent to dropping randomly selected channel-wise attributions in $M$ before collapsing them. Given the calculated channel-wise importance weights $u_c$ and drop probability $p$, the stochastic matching regularizer using Bernoulli masks in framework (2) is formulated as follows:

$$R_c \sim Bern(p), \text{ with } U_c = R_c \cdot u_c \text{ and } u_c = \phi(A_c, F_t, x) \tag{6}$$

where $\phi(\cdot)$ is a function of choice. In this way, we expect that diverse levels of attribute maps of the teacher network are transferred into the student network in the training process. Further diverse strategies can be explored under this setting, such as sharing the drop mask between the teacher network and the compressed network according to their similarity.

## 4   Experiments

In this section, we evaluate the performance of our framework on three distinct methods of compression: unstructured pruning, structured pruning, and knowledge distillation. For the choice of attribution algorithm to evaluate the interpretability of the models, we use Grad-Cam [6] to generate the attribution map for a given data point not only due to its simplicity and popularity but also due to its ability to detect important regions that reflect a model's decision process (Appendix E). For each compression method, we compare the following four methods: naive fine-tuning, equally weighted collapsed activation map matching (3) (denoted as 'EWA'), sensitivity-weighted activation map matching (4) (denoted as 'SWA') and its stochastic version (denoted as 'SSWA'). We apply our matching regularizers on the last convolutional layer of a network. This is justified in the sense that the last convolutional layer conveys the most class distinctive information. For sensitivity-weighted matching and its stochastic variant, we match only the attribution map generated from the top 1 prediction of the full network. This is due to the computational cost of calculating the Jacobian matrix with contemporary automatic differentiation libraries, in which they require separate backpropagation steps for each row of the Jacobian matrix. For the settings described above, we conduct extensive experiments on the Pascal VOC 2012 [28] multi-label classification dataset. Further details on experimental settings and evaluation metrics are provided in Appendix C.

**Evaluating attribution maps**    To the best of our knowledge, there is no commonly agreed metric to measure the deviation of an attribution map to another due to the subjectiveness of attribution algorithms. To assess as objectively as possible, we measure the degree of deformation in attribution maps with cosine similarity, a widespread metric to represent the similarity between two vectors. However, cosine similarity can only measure the directional similarity between two vectors. Thus, the difference in intensity between two attribution maps is not captured. For this cause, we also measure the normalized $\ell_2$ distance between the attribution maps to capture the difference in intensities.

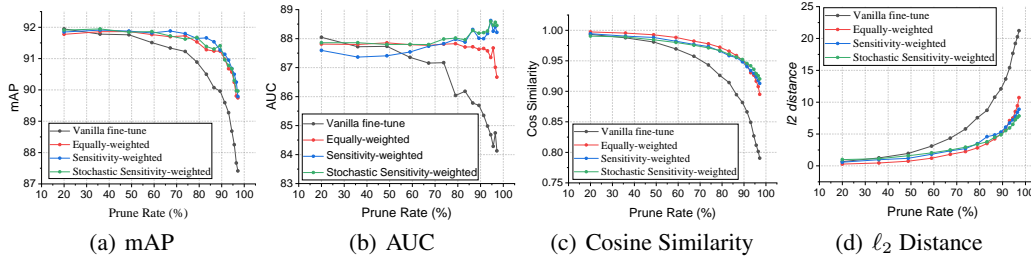

| (a) mAP | (b) AUC | (c) Cosine Similarity | (d) $\ell_2$ Distance |

**Figure 3:** Results of Unstructured pruning experiments on Pascal VOC 2012.

Since samples that the model's prediction is wrong are not 'understood' by the model, their attribution maps are likely to break down. Thus, if we evaluate the attribution performance on the entire test set, models with low predictive performance are naturally at a disadvantage. To compensate for this effect and compare the attributions of all models on the same ground, we only consider the samples that each model *correctly* predicted.

**Comparison with ground truth segmentation labels**  We also evaluate the effectiveness of our framework by comparing the absolute quality of the attribution maps. Towards this, we evaluate the localization capability of the attribution maps by comparing them to ground truth segmentation labels, which is a widely used method to measure the soundness of attribution methods. We utilize the held out 1,449 images with segmentation masks in the PASCAL VOC 2012 dataset. However, the segmentation maps lack the intensity information present in attribution maps. Thus, the heatmaps must be thresholded to be compared. Since the ground truth segmentation labels are imbalanced, the performance of localization is affected by the intensity threshold in which we create the weak localization maps. Thus we compute the ROC-AUC by changing the intensity threshold. We separate the segmentation masks associated with the ground truth labels, calculate the ROC-AUC value for each label, and calculate their average. Moreover, to evaluate how many samples are broken due to compression, we utilize the Point Accuracy [8] that counts whether the max value of the heatmap is inside the segmentation map.

## 4.1 Knowledge Distillation

For our experiments on knowledge distillation, we use the standard network distillation technique introduced in [3]: we train a smaller student model using a linear combination of the typical cross-entropy loss with ground truth label and the KL divergence between the teacher and student output logits. we use the VGG16 network [10] and create smaller student versions of the VGG16 network by maintaining the overall architecture but reducing the number of channels for all layers. We prepare 3 students: one-half (VGG16/2), one-quarter (VGG16/4), and one-eighth (VGG16/8). The teacher network is first initialized with off-the-shelf ImageNet pretrained weights, then trained with the

**Table 2:** Results of knowledge distillation models evaluated against the ground truth (segmentation).

| Network | Method | Prediction Performance | | Attribution Score | |
|---|---|---|---|---|---|
| | | mAP | F1 Score | AUC | Point Acc |
| VGG16 | Teacher | 91.83 | 78.44 | 88.79 | 80.21 |
| VGG16 / 2 | KD | 83.75 | 65.92 | 82.53 | 72.01 |
| | EWA | 86.48 | **68.19** | 85.05 | 80.42 |
| | SWA | **86.56** | 67.78 | 88.12 | 80.66 |
| | SSWA | 86.42 | 67.94 | **88.89** | **81.13** |
| VGG16 / 4 | KD | 81.31 | 62.50 | 80.61 | 68.86 |
| | EWA | 82.46 | 63.57 | 84.18 | 79.34 |
| | SWA | 83.67 | 65.14 | 87.90 | 80.05 |
| | SSWA | **84.47** | **66.13** | **88.10** | **80.26** |
| VGG16 / 8 | KD | 76.91 | 52.51 | 78.74 | 67.26 |
| | EWA | 79.56 | 58.91 | 81.99 | 78.49 |
| | SWA | 80.14 | 60.70 | 87.88 | **79.59** |
| | SSWA | **80.86** | **61.43** | **88.06** | 79.12 |

**Table 3:** Knowledge distillation results measuring the deformation of attribution maps from teacher to student.

| Network | Method | Similarity | |
|---|---|---|---|
| | | Cos | $\ell_2$ |
| VGG16 | Teacher | - | - |
| VGG16 / 2 | KD | 0.705 | 29.84 |
| | EWA | 0.788 | 21.21 |
| | SWA | **0.873** | **12.98** |
| | SSWA | 0.859 | 14.36 |
| VGG16 / 4 | KD | 0.650 | 35.52 |
| | EWA | 0.750 | 25.24 |
| | SWA | **0.841** | **16.23** |
| | SSWA | 0.837 | 16.63 |
| VGG16 / 8 | KD | 0.563 | 44.10 |
| | EWA | 0.652 | 34.90 |
| | SWA | 0.813 | **19.04** |
| | SSWA | **0.842** | 19.49 |

**Table 4:** Unstructured pruning models evaluated against ground truth (segmentation). Among the results of iterative pruning, the last remaining small-est network was evaluated.

|  | Prediction Performance | | Attribution Score | |
|---|---|---|---|---|
| Method | mAP | F1 Score | AUC | Point Acc |
| Full(Teacher) | 91.83 | 78.44 | 88.79 | 80.21 |
| Naive | 87.42 | 70.24 | 84.13 | 75.43 |
| EWA | 89.75 | 74.83 | 86.67 | 79.37 |
| SWA | 89.79 | 75.11 | 88.22 | **79.86** |
| SSWA | **89.96** | **75.51** | **88.45** | 79.25 |

**Table 5:** Unstructured pruning results for attribution map deformation from teacher to student network.

| Method | Cos | $\ell_2$ |
|---|---|---|
| Full(Teacher) | - | - |
| Naive | 0.790 | 21.21 |
| EWA | 0.895 | 10.71 |
| SWA | 0.913 | 8.407 |
| SSWA | **0.920** | **7.826** |

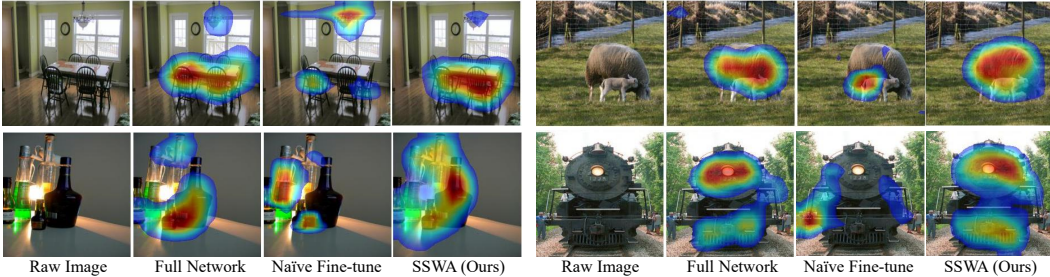

Raw Image    Full Network    Naïve Fine-tune    SSWA (Ours)    Raw Image    Full Network    Naïve Fine-tune    SSWA (Ours)

**Figure 4:** Attribution maps of compressed networks trained with and without attribution matching.

PASCAL VOC 2012 dataset. When the teacher's training is complete, a randomly initialized student is trained with knowledge distillation. In Table 2 and Table 3, we list the results of knowledge distillation experiments. We observe that the network trained with our framework not only effectively preserves the attribution maps, but also consistently outperforms the network distilled without our method in terms of prediction performance, which is measured in *mean-average-precision* (mAP) and F1 score. This result is partly expected from the work [4]. We also observe that matching the sensitivity-weighted activation map outperforms the equally weighted one. We suspect that this gain is caused by the channel weighting scheme and matching an activation map that is conditioned on a class rather than matching a class-degenerated map.

## 4.2 Unstructured Pruning

We evaluate the performance of our method on networks pruned in an unstructured fashion. Unstructured pruning severs individual connections in the network to reduce the number of parameters, resulting in sparse weight matrices. We use the unstructured pruning method proposed in [18] for network pruning. First, we initialize the full VGG16 network with off-the-shelf weights pretrained on ImageNet. Then a full network is trained on the PASCAL VOC 2012 dataset. After the training is complete, the weights of the network are sorted according to their magnitude and a desired amount of weights are pruned. We use pruning rate $\rho_w = 0.2$. After pruning is complete, the remaining sparse network is fine-tuned for 30 epochs on the same dataset. The whole process is then iterated 16 times to produce the final compressed network with pruning rate $\rho = 0.97$. Our matching regularizer was employed at all pruning iterations.

**Table 6:** $\ell_1$-structured pruning models evaluated against ground truth (segmentation).

|  | Prediction Performance | | Attribution Score | |
|---|---|---|---|---|
| Method | mAP | F1 Score | AUC | Point Acc |
| Full(Teacher) | 91.83 | 78.44 | 88.79 | 80.21 |
| Naive | 83.76 | 60.71 | 79.98 | 75.29 |
| EWA | 87.62 | 66.05 | 83.96 | 78.84 |
| SWA | 88.39 | 67.70 | 86.99 | **81.65** |
| SSWA | **89.07** | **68.28** | **88.34** | 81.08 |

**Table 7:** $\ell_1$-structured pruning results for attribution map deformation from teacher to student network.

| Method | Cos | $\ell_2$ |
|---|---|---|
| Full(Teacher) | - | - |
| Naive | 0.764 | 30.04 |
| EWA | 0.855 | 14.74 |
| SWA | **0.911** | **9.102** |
| SSWA | 0.910 | 9.232 |

**Table 8:** Attribution deformation and preservation results on other attribution methods. For this experiment, we use the knowledge distillation with VGG/8. We report the AUC and Point accuracy to evaluate the localization ability of the attribution maps.

| AUC/Point Acc | Grad Cam | Excitation Bp | LRP$_{\alpha=1,\beta=0}$ | RAP |
|---|---|---|---|---|
| Full (Teacher) | 88.79/80.21 | 84.14/74.80 | 85.29/65.48 | 84.54/69.49 |
| Naive | 78.74/67.26 | 76.31/66.31 | 79.60/53.43 | 80.85/56.87 |
| SSWA (Ours) | **88.06/79.12** | **82.31/71.24** | **82.46/64.08** | **83.53/65.66** |

The results of unstructured pruning is presented in Figure 3 and Table 4, 5. Similar to the results in Section 4.1, networks employing matching regularizers exhibit better attribution and predictive performance compared to naive finetuning.

## 4.3 Structured Pruning

For structured pruning, we use the $\ell_1$ structured pruning proposed in [2], in which whole filters are pruned according to the magnitude of each filter's $\ell_1$ norm. The general flow of the experiment is similar to other methods. We use the same ImageNet-initialized VGG16 to train the full network. We use channel pruning rate $\rho_c = 0.7$. For structured pruning, we do not iterate the pruning cycle but execute the process a single time(one-shot pruning), so we set a higher pruning rate. The results of structured pruning experiments are summarized in Table 7 and Table 6. We observe similar tendencies.

## 4.4 Qualitative Evaluation of Attribution Maps

Aside from the quantitative assessment done in previous sections, we also conduct a qualitative assessment of the attribution maps. We draw and examine the attribution maps produced by structure-pruned networks trained with naive fine-tuning and SSWA with respect to the map of the full network. To this cause, we select images among the samples that all the methods have succeeded in predicting the correct label. The images are shown in Figure 4. We observe that even though the predictions of the networks are all correct, the quality of attribution maps produced by the compressed networks with respect to the full network varies. We see that the attribution maps produced by our method most resemble the maps of the teacher network.

## 4.5 Effects on Other Attribution Methods

In the sections above, we observed the effectiveness of our method using Grad-Cam. In this section, in addition to Grad-Cam, we observe how maps produced by other attribution methods are deformed by compression and remedied by our method. We calculate the ROC-AUC curve and point accuracy of other attribution maps including Excitation Backprop [14], LRP [15], and RAP [29] for knowledge distillation with VGG/8. The experimental setting is identical to that of Section 4.1. As described in Table 8, we observe that the maps of the three attribution methods are indeed deformed when compression is performed, and exhibit inferior point accuracy and ROC-AUC performance compared to the network before compression. Moreover, we observe that even though SSWA utilized gradient-based attribution maps akin to Grad-Cam, employing this regularizer helps to preserve other attribution methods including non-differentiable ones [14, 15, 29]. This is partly expected as the decision-critical regions of an input are indeed reflected in Grad-Cam maps (Appendix E). Thus, if any other attribution method is indeed trying to reveal the decisive regions, they are bound to show the regions similar to Grad-Cam.

## 5 Conclusion

In this work, we assert the problem of attribution preservation in compressed deep neural networks based on the observation that compression techniques significantly alters the generated attributions. To this end, we propose our attribution map matching framework which effectively and efficiently enforces the attribution maps of the compressed networks to be the same as those of the full networks. We validate our method through extensive experiments on benchmark datasets. The results show that our framework not only preserves the interpretation of the original networks but also yields significant performance gains over the model without attribution preservation.

## Broader Impact

In the paper, we brought up the attribution deformation problem in compressed networks, and a novel method to combat this issue. As discussed in Section 1, we believe that people trying to deploy deep learning models to safety-critical fields must be aware of this finding to ensure the reliability and trustworthiness of the system. To this end, we may think of a possible scenario. Suppose that a CNN classifier vision module trained with our matching regularizer is utilized in a self-driving system. In case of an accident, we may inspect the records of the deep learning module to learn the decision that caused the accident. In this situation, the model trained with our regularizer will provide more accurate attribution, leading to a cleaner and more just assessment.

However, the sense of attributional safety presented by our method can give a false sense of security and blind trust towards the system and its interpretations, while by no means the system is flawless. For example, a wearable health monitor might predict a person to be healthy, and provide its supporting explanations. If these explanations are blindly trusted, while they are wrong underneath the surface, the user might take reactive measures that are ultimately bad for oneself.

## Acknowledgments and Disclosure of Funding

This work was supported by the National Research Foundation of Korea (NRF) grants (No.2018R1A5A1059921, No.2019R1C1C1009192), Institute of Information & Communications Technology Planning & Evaluation (IITP) grants (No.2017-0-01779, A machine learning and statistical inference framework for explainable artificial intelligence, No.2019-0-01371, Development of brain-inspired AI with human-like intelligence, No.2019-0-00075, Artificial Intelligence Graduate School Program (KAIST)) funded by the Korea government (MSIT). This work is also supported by Samsung Advanced Institute of Technology (SAIT).

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
