[Supplementary Material]

# Supplementary Material: Attribution Preservation in Network Compression for Reliable Network Interpretation

**Geondo Park**[1]*, **June Yong Yang**[1]*, **Sung Ju Hwang**[1,2], **Eunho Yang**[1,2]
KAIST[1], AITRICS[2], South Korea
{geondopark, laoconeth, sjhwang82, eunhoy}@kaist.ac.kr

## A  Deformation of Other Attribution Methods

Here, we observe the deformation of various attribution methods other than Grad-Cam for several compression methods. As in the main paper, we calculate the ROC-AUC curve and the localization accuracy (Point accuracy) of attribution maps including Excitation Backprop [1], LRP [2], and RAP [3]. The AUC denotes the degree of overlap between the ground truth segmentation and the attribution map. Point accuracy [4] is a measure of whether the max value of the heatmap is inside the segmentation map or not. Note that only the samples that the predictions of the network were **correct** are counted for a fair evaluation. As shown in Table 1 and Table 2, we observe that all attribution methods are deformed when compression is performed, and point accuracy and ROC-AUC performance are degraded compared to the scores before compression.

In the main paper, we showed that our attribution matching regularizer partially preserves other non-differential attribution maps even though the matching is executed on the scope of differentiable maps such as Grad-Cam [5]. We leave the task of fully preserving various attribution methods for future work.

**Table 1:** ROC-AUC of four attribution methods on different network compression methods for the PASCAL-VOC dataset.

| ROC-AUC | Params | Grad Cam | Excitation Bp | $LRP_{\alpha=1\beta=0}$ | RAP |
|---|---|---|---|---|---|
| Full Network | 15.22M | 88.79 | 84.14 | 85.29 | 84.54 |
| Knowledge Distillation | 0.29M | 78.74 | 76.31 | 79.60 | 80.85 |
| Structured Pruning | 3.27M | 79.98 | 79.03 | 81.60 | 79.30 |
| Unstructured Pruning | 0.53M | 84.72 | 81.99 | 81.55 | 81.16 |

**Table 2:** Point accuracy of four attribution methods on different network compression methods for the PASCAL-VOC dataset.

| Point Accuracy | Params | Grad Cam | Excitation Bp | $LRP_{\alpha=1,\beta=0}$ | RAP |
|---|---|---|---|---|---|
| Full Network | 15.22M | 80.21 | 74.80 | 65.48 | 69.49 |
| Knowledge Distillation | 0.29M | 67.26 | 66.31 | 53.43 | 56.87 |
| Structured Pruning | 3.27M | 75.29 | 69.22 | 61.13 | 65.01 |
| Unstructured Pruning | 0.53M | 75.43 | 70.28 | 60.23 | 65.26 |

*Equal contribution. Listing order is alphabetical.

# B    Experiments on ImageNet

In addition to the PASCAL VOC 2012 experiments in Section 4 of the main text, we report the results of similar experiments on the ImageNet dataset [6]. The general outline of the experiments is held identical to the PASCAL VOC 2012 experiments except for a few modifications. Since several prior works report that performing knowledge distillation for the ImageNet-1000 classification task is notoriously difficult [7, 8], we omit the distillation experiment and evaluate the performance of our framework on two methods of compression: Unstructured Pruning and Structured Pruning. In section 4, we measured the ROC-AUC of the attribution maps with respect to ground truth segmentation labels. For the following ImageNet experiments, we use the segmentation labels provided by [9]. This data provides ground truth segmentation labels for 4276 images extracted from ImageNet. However, the classification labels of these images do not belong to the ImageNet-1000 task but to the whole ImageNet class labels - the class labels are unusable. Thus, we cannot exclude the scores produced by samples that the models have predicted wrong. We opt for generating the attribution maps of the top-1 prediction of the model for all samples and compare it to the ground truth segmentation labels.

## B.1    Unstructured Pruning

We conduct experiments on unstructured pruning [10]. For this experiment, we use the one-shot pruning pipeline instead of iterative pruning due to the computational cost of repeatedly fine-tuning on ImageNet. In the fine-tuning phase, the pruned network is fine-tuned for 10 epochs with batch size 180. We report on two pruning rates of $0.6$ and $0.9$. For both cases, we observe that our method better preserves the attribution maps compared to the naive compressed network (Table 3). However, the number gaps for all metrics are smaller compared to the PASCAL VOC 2012 experiment. We suspect that this is due to the relative easiness of the ImageNet in terms of localizing. For most ImageNet samples, a single main object is centered on the image. This implies that in most cases the network only has to focus on the center part of the image. Thus, the network only has to maintain its focus on the center part of the image when it is compressed, which is a relatively easy task.

**Table 3:** Results of unstructured pruning on ImageNet.

| Prune Ratio | Method | Predictive Performance Top-1 Acc | Attribution Score AUC | Point Acc | Attribution Similarity Cos | $\ell_2$ $(10^{-5})$ |
|---|---|---|---|---|---|---|
| Full Network | - | 73.37 | 81.64 | 91.90 | - | - |
| 60% | Naive | 73.31 | 76.01 | 91.21 | 0.975 | 1.614 |
| | EWA | 73.33 | 76.52 | 91.63 | 0.977 | 1.603 |
| | SWA | **73.36** | 79.82 | 91.67 | 0.980 | 1.26 |
| | SSWA | 73.32 | **80.88** | **91.78** | **0.981** | **1.206** |
| 90% | Naive | 70.38 | 75.43 | 90.39 | 0.925 | 4.80 |
| | EWA | **70.52** | 75.68 | 90.75 | 0.919 | 5.18 |
| | SWA | 70.48 | 79.85 | **91.35** | **0.939** | 3.88 |
| | SSWA | 70.46 | **80.63** | 90.93 | **0.939** | **3.87** |

## B.2    Structured Pruning

We conduct experiments for structured pruning methods on ImageNet. For these experiments, we use ResNet34 instead of VGG16 due to computational constraints. We prune the network with the channel pruning rate set to $\rho_c = 0.1$ due to the difficulty of the ImageNet classification task. After pruning, the network is fine-tuned for 20 epochs. We observe same tendencies in the results (Table 4). Our method outperforms naive compression in terms of maintaining the attribution maps.

**Table 4:** Results of $\ell_1$-structured pruning on ImageNet.

| Method | Predictive Performance Top-1 Acc | Attribution Score AUC | Point Acc | Attribution Similarity Cos | $\ell_2$ $(10^{-5})$ |
|---|---|---|---|---|---|
| Naive | 70.06 | 81.20 | 83.96 | 0.982 | 2.248 |
| SWA | 70.102 | **84.70** | 88.33 | **0.988** | 1.550 |
| SSWA | **70.486** | 84.65 | **88.51** | **0.988** | **1.521** |

# C  Experimental Details For the PASCAL VOC 2012 Experiments

## C.1  Dataset

We used the Pascal VOC 2012 [11] multi-label classification dataset which consists of 5717 training and 5823 validation high-resolution images. Among the validation samples, we utilize 1,449 held out images with segmentation masks for localization evaluation. The dataset can be downloaded from the following link: http://host.robots.ox.ac.uk/pascal/VOC/voc2012/ .

We normalize the input with mean $[0.4589, 0.4355, 0.4032]$ and standard deviation $[0.2239, 0.2186, 0.2206]$. For data augmentation, we use random resized crop and random horizontal flip provided by Torchvision and Pytorch. [12].

## C.2  Training

**Hyperparameters.**   For the CNN implementation, we used the vgg16_bn implementation provided by Torchvision. To train the full network(teacher), we used stochastic gradient descent (SGD) with learning rate 0.1, momentum 0.9, weight decay of $5 \times 10^{-4}$. We trained the model with batch size 128 for 250 epochs. For distillation experiments, we used SGD with learning rate 0.1, momentum 0.9, and weight decay $10^{-4}$. We trained the models for 350 epochs with batch size 64. For unstructured pruning, we used SGD with learning rate $10^{-3}$, momentum 0.9, and weight decay $10^{-4}$. We trained the models for 16 pruning iterations where a single iteration is of 30 epochs. A batch size of 64 was used. For structured pruning, a one-shot pruning scheme of 60 epochs was used. The optimizer hyperparameters and batch size are identical to unstructured pruning. We used regularizer strength of 100 for EWA and 50 for SWA and SSWA, across all compression methods.

**Apparatus and Runtime.**   Our experiments on PASCAL took around 100 seconds per epoch on a single machine equipped with 2 Intel(R) Xeon(R) CPU E5-2630 v4 CPUs and 4 NVIDIA Geforce TITAN Xp graphics cards.

## C.3  Evaluation

Given a pair of attribution maps from before $(M_t)$ and after $(M_s)$ compression, the cosine similarity is computed as follows:

$$\cos(\theta) = \frac{M_t * M_s}{\|M_s\|\|M_s\|}.$$

The normalized $\ell_2$ distance between the attribution maps are evaluated as follows:

$$\ell_2 \text{ distance} = \left\| \frac{M_s}{\|M_s\|_2} - \frac{M_t}{\|M_t\|_2} \right\|_2.$$

To evaluate against ground truth segmentation labels, we use ROC-AUC and point accuracy provided by the pointing game [4]. Since segmentation labels are provided as 0's and 1's, it is possible to evaluate the quality of attribution maps as a binary classification task. In this sense, we normalize the attribution maps to take values within $[0, 1]$ interval and apply a decision threshold to record the accuracy. This process can be repeated with different thresholds to produce a ROC curve. Using this curve, we report the AUC of the ROC curve. The pointing game accuracy is measured in the following manner: if the spatial location of the maximum value of an attribution map is located within the segmentation mask, it is a hit. Otherwise, it is a miss. This process is repeated and averaged for the test samples.

# D  More Examples

Below, we provide visualizations of attribution maps for additional samples for extended qualitative assessment.

| Raw Image | Full Network | KD | Structured Pruning | Unstructured Pruning | KD with Ours |

**Figure 1:** Attribution maps of a network before and after network compression. These figures are examples that the networks are predicting the correct label (airplane, sofa, cat, bird, airplane, cat, person, person, airplane, bottle) before and after compression but produce different attribution maps. The last column of examples comes from the network trained with knowledge distillation and our regularization. The results show that our regularization indeed preserves attribution maps.

**Figure 2:** Attribution maps of compressed networks with structured pruning trained with and without attribution preservation regularization. These examples also predict the correct label (person, horse, cow, train, bus, cat). Examples show that our approach preserves attribution maps.

**Figure 3:** Evaluation of ROAR on knowledge distilled vgg16/4 network measured with a vgg-11 classifier. Results show that the Grad-Cam maps are significantly better at attributing than the random baseline. Also, we see that the network trained with our method achieves almost equal attribution performance in terms of ROAR.

# E   Validation of Grad-Cam Maps as a Mean to Measure Attribution Quality

Here, we conduct additional experiments to ascertain Grad-Cam's capability to extract regions that are deemed important by the model. We additionally measure the perturbation metric, *RemOve-And-Retrain* (ROAR) [13], to evaluate how well the attribution maps from compressed networks explain the model behavior. To measure ROAR, attribution maps for the entire training data are extracted from the network undergoing the test. Then, the top-$k$ pixels of an image ranked by the attribution map is removed. Finally, a separate classifier is retrained on this perturbed dataset. If the attribution map was to accurately represent the importance of the pixels, the classifier must exhibit lower predictive performance. We measure this metric on the full network, naively distilled network, and a network trained with our method. Random attribution was compared as a baseline. **(a)** As shown in Figure 3, all Grad-Cam perturbations (from different models) were able to lower the F1 score more than random perturbations, which verifies that Grad-Cam indeed reflects a model's decision-making process. **(b)** The student trained with our method scored almost on par with the full network. This indicates that the attributions (which reflect a model's decision process) are indeed preserved by our method.