[Reviews · NeurIPS 2020]

Review 1

Summary and Contributions: Update based on author feedback: The authors addressed both of my main concerns, and so, as promised, I am raising my score. On the subject of novelty: 1) I agree with other reviewers that the similarities to Zagoruyko et al. reduce novelty (particularly since R2 pointed out that they failed to clarify that Zagoruyko et al. also included sensitivity-based regularizers). However, in Zagoruyko et al. they seem to have recommended the activation-based regularization as the preferred method (they wrote "we also trained a network with activation-based AT in the same training conditions, which resulted in the best performance among all methods"). Thus, I think it is meaningful that these authors have demonstrated that a sensitivity-based approach works better. 2) As R2 noted based on the author response, a key difference between the approach in this work and the approach in Zagoruyko et al. is that Zagoruyko et al. takes the gradient w.r.t. the loss but in this work they take the gradient w.r.t. an output logit. In addition to losing task-specific information, the gradient of a categorical cross-entropy loss tends to zero for examples that are predicted confidently & correctly - thus, taking the gradient w.r.t. the loss would focus on examples where the model's predictions are most *incorrect* (those examples would both have the largest loss and the largest loss gradients); by contrast, taking the gradient w.r.t. the task logit (as these authors do) would avoid both those issues. I think this difference is nontrivial. 3) I somewhat disagree with the line in the author response re. Zagoruyko et al. where they say "since they only match the gradients at the input level, the information in the intermediate layers (and thus their decision processes) are not appropriately transferred" because the idea of matching at higher layers is very explicitly acknowledged in Zagoruyko et al., who write "in this work we consider only gradients w.r.t. the input layer, but in general one might have the proposed attention transfer and symmetry constraints w.r.t. higher layers of the network." 4) I agree with the authors of the present work that the matter of framing is important; in Zagoruyko et al., they focused specifically on improving performance. As many reviewers noted, this work seems to be the first to point out that even when performance is retained, the attribution maps can become distorted during compression. That in itself is a valuable observation; without work like this, people may not have thought to perform attribution regularization if they felt that the performance was fine. Ideally, the authors would compare to the sensitivity-based regularizer from Zagoruyko et al. and show that their proposed approach works better. However, given that the authors of Zagoruyko et al. recommended the activation-based regularizer as the best, and that the authors of the present work showed that they outperformed the activation-based regularizer, I personally feel that the work is above the acceptance threshold. ------------- The paper introduces the problem that a compressed model may have distorted attributions relative to the parent model. The authors propose a solution to this problem wherein the attributions of the child model are regularized to match those of the parent model. Using image segmentation for ground-truth explanations, they empirically verify that, across three different compression strategies (distillation, structured pruning and unstructured pruning), the resulting compressed models both resemble the parent model in terms of the attributions and also have higher-quality attributions compared to naively-compressed models.

Strengths: - To my knowledge, the problem formulation is novel; I don't believe other works have specifically investigated the issue of distorted attributions in compressed models. - The methodology makes sense, and the empirical results seem convincing in terms of matching the attributions to the parent model and also obtaining higher-quality attributions w.r.t. the image segmentation ground truth.

Weaknesses: (1) My primary concern is the that while the authors have demonstrated that models compressed using their proposed strategy obtain better results with respect to quantifying the quality of the attributions w.r.t. a ground-truth segmentation task, the authors stated that "if we evaluate the attribution performance on the entire test set, models with low predictive performance are naturally in a disadvantage. To compensate for this effect and compare the attributions of all models on the same ground truth, we only consider the samples that each model predicted correctly". When compressing models, we *do* also care about whether the distilled model retains good predictive performance, and from the tables/figures in the main text I did not get a sense of whether the **predictions** of the models distilled using the proposed method are also more accurate (the figures and tables all appear to be reporting measure of attribution performance, not of prediction performance). The authors have listed the trustworthiness of a model as a reason for using the proposed method - the trustworthiness of a model also depends on having reliable predictions, so I think it is essential to report this as well. I noted that Tables 3 and 4 in the supplement have a column labeled "accuracy" that shows that the accuracies of SWA & SSWA are not lower relative to "Naive" - can the authors confirm whether "accuracy" in these columns refers to prediction accuracy? If so, could the authors report a similar measure of prediction performance for the tables in the main text? **If the authors can confirm that predictive performance is still maintained when compared to the naive compression strategy, I would be willing to revise my score towards acceptance.** (2) (This is a more minor concern as I don't expect it to be likely, but I think it is still worth thinking about) The authors evaluated the saliency maps w.r.t. (a) whether the saliency maps of the child model resemble those of the parent models, and (b) whether the saliency maps match an image segmentation ground-truth - however, I did not see a measure of whether the saliency maps are **indicative of the decision-making process of the child model**. It has been shown before that saliency maps are "fragile" in that they are susceptible to adversarial perturbations (https://arxiv.org/abs/1710.10547), so it is conceivable to me that a child model can generate a saliency map (via a method like Grad-CAM) that superficially mimics the saliency map of the parent model but which is not reflective of the child model's decision-making process (particularly because Grad-CAM relies on gradients, which don't incorporate saturation effects as discussed in the DeepLIFT/IntegratedGradients papers). In such a situation, the saliency maps would be **misleading** with respect to the child model. I do not expect it to be the case that this is happening, but I think it is worth addressing for completeness. The perturbation metric proposed in the FullGrad paper, where the *least important* pixels are perturbed and the change in the model performance is quantified, may be a good way to measure the extent to which the saliency map is faithful to the child model's decision-making process: https://arxiv.org/abs/1905.00780

Correctness: My main concern about the methodology, as described in the previous section, is that the child model may be superficially mimicking the saliency map of the parent model without actually adopting the parent model's decision-making process (particularly of a "local" explanation method like Grad-CAM is used). Perturbation experiments could address this concern. However, I do not expect this phenomenon to be likely, so it is not a very major concern. Another small concern: in line 193, the authors describe applying a ReLU operation to the channel importance obtained from Grad-CAM. In general, discarding negative gradients (as is done in, e.g., Guided Backprop and DeconvNet) has been shown to diminish the quality of the attributions (e.g. by making them prone to failing sanity checks; https://arxiv.org/abs/1912.09818). I am thus somewhat concerned that the authors felt the need to discard negative channel importance here, because negative importance can still be relevant for classification.

Clarity: For the most part, yes. One piece I was unclear on was whether only correctly-predicted examples from the parent model were used for regularization during training (the authors wrote that "we only consider the samples that each model correctly predicted" in the context of the test set because those are the attributions that are likely to be reliable; I was unsure whether this was also leveraged during training).

Relation to Prior Work: To my knowledge, yes (I am not very familiar with the model compression literature, hence my lower confidence rating).

Reproducibility: Yes

Additional Feedback: I have mentioned some suggestions under "Weaknesses"; what's listed here are more minor issues: (1) Assuming that the Grad-CAM backpropagation was started w.r.t. the logits of the softmax layer, it may be a good idea to normalize the logits such that the mean across all classes sums to 0. Normalizing the logits of a softmax does not change the output of a softmax, but it would change the attributions (in that, if a particular channel has the same contribution to all softmax logits, it is effectively contributing to none of the softmax logits). This is also mentioned in the section "Adjustments for softmax layers" in the DeepLIFT paper: https://arxiv.org/pdf/1704.02685.pdf (2) I think it is worth reflecting on the extent to which image segmentation is a good "ground truth" explanation, because I think background pixels can often be relevant for a class prediction (for example, if the background is green, then a prediction of "cow" is more likely than if the background is pink). That said, I agree that, broadly speaking, the "pointing game" measure is likely valid (i.e. the peak attribution should fall within the segmented region). (3) (minor) I would be curious how Grad-CAM (which averages the gradients over a channel) performs relative to simply doing "activation*gradient" at each individual neuron in the convolutional layer.


Review 2

Summary and Contributions: The paper starts from the observation that compressed networks can produce attribution maps significantly different from the corresponding original uncompressed networks, despite having comparable accuracy. The authors argue that this is problematic, as similar accuracy does not necessarily mean that the two networks process information in the same way. They propose an attribution-based regularization term to steer the fine-tuning towards local minima that have both high predictive accuracy and good matching of attributions between the original and the compressed network.

Strengths: As neural networks will be increasingly used in safety-critical domains, the problem of understanding how they process input information is important. To the best of my knowledge, the observation that compression techniques might shift the attention of the network towards less relevant input features, despite preserving the model accuracy, is novel and therefore potentially relevant for the XAI and security community. The authors show empirically on VOC and ImageNet that it is possible to mitigate the problem of "attribution shift" employing attributions as regularization term, and that this often produces better results than simple activation matching as proposed by Zagoruyko et al. 2017. The paper and the proposed method are easy to understand. The proposed regularization technique is based on a well-known attribution method and easy to implement. The framework can be readily applied to several compression techniques, such as structured/unstructured pruning and KD.

Weaknesses: I have two main concerns regarding this paper: 1) as the goal is to match the attributions between two networks, the idea of adding an attribution-based regularization term in the cost functions seems a trivial and straight-forward solution to me. Moreover, a very similar regularization term was previously proposed by Zagoruyko et al. 2017 who investigated not only activation matching but also the use of gradient-based attributions (in particular, sensitivity map by Simonyan et al. ). While Zagoruyko et al. formulated the problem as "attention transfer", practically their motivation was the same: ensuring that a student model "pays attention" to the same features as the parent. Although it is true that they were only interested in improving the network accuracy, I believe this paper does not add a significant contribution to the method. The authors suggest the use of Grad-Cam as an attribution method but there is no theoretical nor empirical evidence that this method provides superior results than sensitivity map as suggested by Zagoruyko et al. or other gradient-based attribution methods (Gradient x Input, Integrated Gradients, DeepLIFT, or others). Finally, Stochastic matching does not seem to find a theoretical justification. What is the rationale for dropping randomly selected channels and why this should work better than using actual attributions? The connection to dropout is not clear to me. 2) the experimental section does not provide error bounds. As the performance gap between the different methods seems marginal (in particular between EWA and (S)SWA), I wonder if the difference is significant at all. The results might be affected by some stochasticity in the pipeline (e.g. SGD and channel sampling in SSWA). Without providing any standard deviation for these results across different runs, it is impossible to assess the significance of the results.

Correctness: Some claims require clarification. In particular, the connection between Stochastic matching and dropout. Claims such as "preserves the interpretation of the original networks" and "signi´Čücant performance gains" cannot be assessed without error bounds in the experimental section.

Clarity: While the paper is somehow understandable, I have the feeling that the paper would benefit from professional proofreading as several sentences sound odd to me (as a non-native speaker).

Relation to Prior Work: The paper should better explain what is inherited from Zagoruyko et al. as, currently, it seems that Zagoruyko et al. only investigated equally weighted activation matching, while actually they also investigated sensitivity-based regularizers. There is also a line of works [1-3] that investigated training neural networks using attribution as regularizers. The authors might want to compare and contrast with these works. [1] https://arxiv.org/abs/1703.03717 [2] https://arxiv.org/abs/1906.10670 [3] https://arxiv.org/pdf/1909.13584.pdf

Reproducibility: Yes

Additional Feedback: Due to the lack of novelty in the method and empirical results that are not particularly strong, I believe this paper requires some more work. However, I still believe that both the motivation of the paper as well as the observation of the attribution-shift phenomenon during compression are relevant. I would suggest the authors do some more in-depth analysis of the phenomenon. This could be, for example, investigating some of the following: why it occurs in the first place, show attributions computed by other methods (is the problem evident with other methods than GradCAM?), discuss (possibly from a theoretical point of view) how can attributions be so different despite the accuracy being similar (are the logits preserved? What about the activation of the hidden layers?), discuss possible robustness implications (if attributions look wrong despite high accuracy, does it mean that the network is actually less robust and more sensitive to wrong areas of the input? this could be investigated with ablation tests). Some minor comments: - Table 1: "AUC" is not defined. Even if a person understands that this is the area under the curve, it is not clear which are the dimensions that define the curve until the reader reaches page 7. The authors might consider making the caption of the figure more explicit. - l. 166: "V() is a rectification function" - does this mean V() is a ReLU? Probably not as in (3) it is chosen to be the identity. The authors might want to change the definition of V() to avoid confusion. - it is not clear to me whether the weight U of stochastic matching is purely based on the Bernoulli distribution (as it seems from the first section of page 6) or whether instead the stochasticity is added on top of the gradient-based weights obtained by (4) as line 223 seems to suggest ("its stochastic version"). I believe it is the former, but then why call it SSWA? - page 7: "mAP" used in Figure 3 and other tables is nowhere defined. - l. 277: the full network is trained from scratch or only fine-tuned? - there is often a missing whitespace before an opened bracket "(", e.g. line 290) ============================================== I increase my score after reading the author response. The newly provided results with deviation and the clarification about the differences with Zagoruyko et al. are convincing. On the other hand, the novely of the method remains limited and I believe this could be a much stronger contribution with a discussion/comparison of other gradient-based attribution methods and with the loss function used by Zagoruyko et al. I still believe that a more in-depth analysis of the phenomenon of attribution shift (with some open questions mentioned above) would be very interesting and could make the work stronger.


Review 3

Summary and Contributions: This paper highlights the surprising fact that network compression, while maintaining the accuracy of the original network, changes the regions of attention of the network, making it less explainable. This is addressed by introducing a regularization term that encourages the attribution maps of the student network to match that of the teacher one.

Strengths: - As far as I am aware of, this paper is the first work to notice that the regions on which a network focuses are affected by compression/distillation. I find this surprising and interesting. - The experiments demonstrate convincingly that the proposed SWA regularizer addresses this issue. - The paper is clearly written and could be relatively easily reproduced (let alone the fact that the code is provided).

Weaknesses: Technical novelty: As acknowledged by the authors, [4] proposed a very similar regularizer (see Eq. 2 in [4]). In fact, the form of Eq. 2 in [4] is quite general, as any function F() could potentially be used. In practice, the authors of [4] studied several functions, i.e., not only the one referred to as EWA in this submission, although this one was the best-performing one in [4]. Altogether, I acknowledge that the motivation behind [4] was different from the one here and that the proposed formulation is somewhat more general and more effective than the one in [4]. However, I feel that the technical novelty remains on the weak side. Presentation: While the paper is clearly written, it could benefit from some additional analysis. In particular: - As mentioned above, I do appreciate the interest of observing that attribution maps are affected by compression. However, I feel that the authors fail to study and explain why this happens. In particular, in the context of pruning, I find particularly surprising that the fine-tuning stage does not address this issue. I would be glad to hear some hypothetical explanations from the authors. - What is the motivation behind the rectification function V()? Why does one need it, and why is a ReLU an appropriate choice (better than alternatives)? - What is the motivation behind the stochastic matching approach? Experiments: The experiments are in general convincing. However: - It would be interesting to study the sensitivity to \beta. - The additional results on ImageNet in the supplementary material (Table 3) show that the compressed networks have a higher AUC than the full network. Can the authors explain this? #### POST-REBUTTAL COMMENTS #### I would like to thank the authors for their responses. I acknowledge that there are some differences w.r.t. [4]. However, I still feel that the similarities make the novelty on the weak side for NeurIPS. Furthermore, while the rebuttal indeed clarifies a few points, others remain unclear, such as why fine-tuning post-pruning doesn't solve the problem by itself, the motivation behind the function V(.) and the influence of \beta. Therefore, while this paper is essentially borderline, I tend to remain slightly on the rejection side.

Correctness: The claims and methodology are correct.

Clarity: The paper is clearly written, but one point nonetheless bothers me: At the beginning of Section 4, the authors mention that Grad-Cam is used to generate the attribution maps. However, it seems to me that these maps depend on the regularizer used, i.e., they are generated using Eq. 3 for EWA, using Eq. 4 for SWA, and using the stochastic variant for SSWA. Is Grad-Cam used for some other purpose?

Relation to Prior Work: The relation to prior work is acknowledged, although, as discussed above, the technical novelty over [4] is limited.

Reproducibility: Yes

Additional Feedback: - Strictly speaking, experimental evaluation is not a contribution and should thus not be listed as such in the introduction. - In unstructured pruning, is the regularizer used in every fine-tuning step?


Review 4

Summary and Contributions: This paper aims to compress the neural network by preserving visual attribution. The authors observe that existing network compression methods only focus on simulating the performance of the target network, so their attribution does not match that of the target network. An attribution-aware compression method is proposed and evaluated on PASCAL VOC 2012 and ImageNet datasets, under several network compression techniques; structured pruning, unstructured pruning, and knowledge distillation.

Strengths: + This paper is well-written and easy to follow. + It is interesting to find that the existing network compression methods do not preserve the attribution map, and the method to address the problem is well-motivated. + Evaluation is done on several network compression techniques and several datasets.

Weaknesses: - Novelty: Finding that existing network compression methods do not preserve attributions is interesting, but this problem has already been partially addressed in [4]. Even [4] considers gradient-based attention (with respect to input image x). - Differentiable attribution methods: Conditions requiring differentiable attribution methods seems not trivial. Methods that do not use gradients seem difficult to apply. For examples, Fong et al., Interpretable explanations of black boxes by meaningful perturbation. Fong et al., Understanding deep networks via extremal perturbations and smooth masks. Schulz et al., Restricting the Flow: Information Bottlenecks for Attribution. Chang et al., Explaining Image Classifiers by Counterfactual Generation. - Generalization for different attribution methods: The experiment of generalization for different attribution methods is missing. It would be interesting to add experiments on how well a network, trained by regularization that preserves attribution maps by Grad-CAM, can preserve the attribution obtained by other attribution methods. Especially, if some methods cannot be applied to this method (because of the differentiability), pleas verify that the interpretations from those (which are not differentiable) can be preserved by the proposed method trained with Grad-CAM.

Correctness: Their findings are reasonable and well visualized. The method was proposed to solve the problems, and evaluated appropriately.

Clarity: This paper is well-written and easy to follow.

Relation to Prior Work: 1. It is necessary to analyze more attribution methods and study whether they can be used in this method.

Reproducibility: Yes

Additional Feedback: Minor comments: (1) Results on ImageNet dataset are shown in supplementary material, but the reviewer thinks it would be better to include them in the main paper. (2) Among raw images in Figure 1, only the third image contains white contour. (3) In Table 3 (supplementary) prune ratio 60%, EWA has higher point accuracy than SSWA, but the result of SSWA is mistakenly bolded. (4) It would be interesting to show the results (in Tables 2 and 3) with lightweight networks trained from scratch to observe the localization ability of those networks. ======= Post author feedback ======= Thanks a lot for the authors' reply. I have read all the comments from other reviewers and the author feedback, and I would keep my original rating. (1) I still feel that novelty is limited. It was modified to be minor compared to Zagoruyko et al. This paper provides a discussion different from the existing method, but considering the high standard of NeurIPS, I think this modification is not sufficient. (2) I was concerned about whether non-differentiable attribution methods other than gradient-based methods could be applied, but these were not addressed in the rebuttal. The generalization experiment of rebuttal is also performed only in gradient methods.

[Author Response · NeurIPS 2020]

We thank the reviewers for their constructive feedback. We are encouraged that all reviewers found our observation
and motivation - attribution map distortion while compressing networks - to be novel and interesting [**R1,R2,R3,R4**]
and our approach to be intuitive [**R1**], readily applied [**R2**], and evaluated with convincing experiments [**R1,R3**]. We
address the reviewer comments as much as space permits and will include all feedback in the final version.

[**R1,R2**] **Evaluation of predictive performance and at-**
**tribution score with** *Error Bound*. We reported the (multi-
label) predictive performance measure in main the text with
*mean-average-precision (mAP)*. We apologize for not elab-
orating this [**R1,R2**]. Also, "accuracy" in Appendix table
3,4 refers to the ImageNet top-1 accuracy [**R1**]. Moreover,

| Method | Predictive Performance | | Attribution Score | |
|---|---|---|---|---|
| | mAP | F1 Score | AUC | Point Acc |
| Full (Teacher) | 91.79±0.16 | 78.44±0.23 | 88.68±0.17 | 80.16±0.16 |
| Naive-KD | 81.44±0.19 | 62.71±0.28 | 80.51±0.25 | 69.18±0.17 |
| EWA | 82.31±0.22 | 63.16±0.31 | 84.23±0.22 | 79.19±0.22 |
| SWA | 83.63±0.19 | 64.94±0.24 | 87.73±0.19 | 79.91±0.21 |
| SSWA | **84.37±0.26** | **66.27±0.35** | **88.13±0.22** | **80.14±0.24** |

we will add one more predictive performance measure: F1 score. Following the [**R2**]'s comments, we additionally
report the performances with standard variation through the 5 same experiments to be more convincing results. In above
the table, we report the predictive performances and attribution scores of various methods including our methods for
knowledge distillation with vgg/4. For other compression cases, we will incorporate in the final version.

[**R1,R2,R3,R4**] **Other evaluation of attribution map & Reliability of Grad-Cam.** Fol-
lowing **R1**'s suggestion, we will add the perturbation metric, RemOve And Retrain
(ROAR) [A], to evaluate the how well attribution maps from compressed networks explain
the model behavior. This test is as follows: remove the top-$k$ pixels of an image ranked
by the attribution map for the entire training data, and retrain a classifier on this perturbed
dataset. If the attribution map accurately represents the importance of the pixels, the
classifier must have lower predictive performance. Here, we show a quick experiment
using vgg-11 classifier. **(a)** In the right graph, all Grad-Cam perturbations (from different

models) were able to lower the F1 score more than random perturbations, which verifies that Grad-Cam indeed reflects
a model's decision making process. **(b)** The student trained with our method scored almost in par with the full network.
This indicates that the attributions (which reflect a model's decision process) are indeed preserved by our method.

[**R4**] **Preserving differentiable attribution map also preserves**
**other attributions.** We observed the deformation of various attribu-
tion maps in Appendix A. Following **R4**'s suggestion, we evaluated

| AUC/PointAcc | Excitation Bp | LRP$_{\alpha=1\beta=0}$ | RAP |
|---|---|---|---|
| Full (Teacher) | 84.2/74.8 | 85.3/65.5 | 84.5/69.5 |
| Naive-KD | 76.3/66.3 | 79.6/53.4 | 80.9/56.9 |
| SSWA (Ours) | **82.3/71.2** | **82.5/64.1** | **83.5/65.7** |

other attribution maps for the model using our method. We observe that it also helps preserving other attribution maps.
Since most attribution maps [14, 15] are generated with gradients and activations, preserving one may help others.

[**R2,R3,R4**] **Technical novelty.** In [4], the authors introduce attention map transfer and gradient transfer. Although
our method and theirs share certain aspects, we believe that there are enough differences to them. **(a)** The problem of
focus is distinct. [4] focuses on boosting the *predictive performance* of a student network, while our method focuses
on preserving the *attribution* power of a network while being compressed. **(b)** The function of the loss functions
are different. Attribution map used in attention matching in [4] lacks label-specific attribution information since all
activation maps are equally weighted and aggregated. In other words, this form of attribution map may tell where the
network is looking, but it holds no "meaning". Thus, this regularizer may teach the student how to look and distinguish
objects, but does not pass on the information of "what" and "why" it should look at a certain region. Gradient matching
of [4] does hold some label-specific information. However, since they only match the gradients at the input level, the
information in the intermediate layers (and thus their decision processes) are not appropriately transferred while our
method is able to transfer information from intermediate layers by collapsing the channels.

[**R2,R3**] **Motivation of stochastic matching.** The intent behind our stochastic matching regularizer is to facilitate
the transfer of relevant information and prevent overfitting. Several recent works utilize this concept to boost the
performance of knowledge transfer between a teacher and a student. In [23], they encourage the teacher-student
information transfer by using a gaussian dropout to maximize the mutual information between teacher and student
features. Injection of stochasticity can also be found on other similar fields such as continual learning [B] and domain
adaptation [C]. Based on the empirical evidences of performance increase presented by the literature mentioned above,
we believe that it is plausible to motivate our stochastic matching method.

[**R3**] **Hypothetical explanation for the attribution deformation problem.** As a network is compressed, it must cram
its decision procedures(information) inside a smaller memory. If so, it would be unable to use "standard" decision
procedures, but must resort to using shortcuts and inklings that means less to humans. Thus, its decision procedures
would become harder to interpret which is reflected in its distorted attribution map.

**Clarity and minor details.** [**R2**] Is the "full net" finetuned or trained from scratch?: *Fine-tuned.* [**R3**]··· is Grad-Cam
used for some other purpose?: *the qualities of Grad-CAM maps extracted from EWA, SWA, SSWA were evaluated.* [**R3**]
We apologize for the mis-reporting: In Appendix Table3, the 'Full Network''s AUC should be changed: *75.92 → 81.64*

[A] Hooker, Sara, et al. "A benchmark for interpretability methods in deep neural networks." NeurIPS. 2019.; [B] Lee, et al. "Overcoming Catastrophic Forgetting by Incremental Moment Matching", NeurIPS. 2017.; [C] Saito, et al. Adversarial Dropout Regularization, ICLR. 2018.;


[Meta-Review · NeurIPS 2020]

I recommend this paper to be accepted. The proposed method is simple and effective. Although I disagree with some specific choices that the authors made (e.g. the choice of the attribution method, it would be a lot more interesting to use masking methods e.g. Dabkowski and Gal 2017 or Zolna, Geras and Cho 2020, which are a lot stronger), I think the general idea of fixing the attribution in the compressed network is sufficiently interesting that this paper could be appreciated at NeurIPS. Having said that, I strongly encourage the authors to take into account the input of the reviewers and improve the camera-ready version of the paper.